

# Adult zebra finches rehearse highly variable song patterns during sleep

Brent K. Young[1], Gabriel B. Mindlin[2], Ezequiel Arneodo[2] and Franz Goller[1]

[1] Department of Biology, University of Utah, Salt Lake City, UT, United States of America
[2] Physics Department, University of Buenos Aires, Buenos Aires, Argentina

## ABSTRACT

Brain activity during sleep is fairly ubiquitous and the best studied possible function is a role in memory consolidation, including motor memory. One suggested mechanism of how neural activity effects these benefits is through reactivation of neurons in patterns resembling those of the preceding experience. The specific patterns of motor activation replayed during sleep are largely unknown for any system. Brain areas devoted to song production in the songbird brain exhibit spontaneous song-like activity during sleep, but single cell neural recordings did not permit detection of the specific song patterns. We have now discovered that this sleep activation can be detected in the muscles of the vocal organ, thus providing a unique window into song-related brain activity at night. We show that male zebra finches (*Taeniopygia guttata*) frequently exhibit spontaneous song-like activity during the night, but that the fictive song patterns are highly variable and uncoordinated compared to the highly stereotyped day-time song production. This substantial variability is not consistent with the idea that night-time activity replays day-time experiences for consolidation. Although the function of this frequent activation is unknown, it may represent a mechanism for exploring motor space or serve to generate internal error signals that help maintain the high stereotypy of day-time song. In any case, the described activity supports the emerging insight that brain activity during sleep may serve a variety of functions.

## INTRODUCTION

Brain activity during sleep is thought to play an important role in consolidation of declarative and procedural memory (e.g., *Maquet, 2001*; *Stickgold, 2005*; *Diekelmann & Born, 2010*; *Yang et al., 2014*; *Tononi & Cirelli, 2014*), and one suggested mechanism of how neural activity effects these benefits is through reactivation of neurons in patterns that resemble those of the preceding experience during the active period (*Wilson & McNaughton, 1994*; *Dave & Margoliash, 2000*; *Hahnloser, Kozhevnikov & Fee, 2002*; *Hahnloser, Kozhevnikov & Fee, 2006*; *Euston, Tatsuno & McNaughton, 2007*; *Peyrache et al., 2009*; *Shank & Margoliash, 2009*). Specifically, motor replay during sleep can be induced by sensory cues and is thought to consolidate motor memory through sensorimotor re-processing in the relevant brain areas (*Derégnaucourt et al., 2005*; *Margoliash, 2005*; *Orban et al., 2006*; *Hupbach et al., 2009*). However, the role of sleep on memory formation for a specific newly acquired motor skill is debated (e.g., *Korman et al., 2003*; *Rickard et al., 2008*; *Brawn, Nusbaum & Margoliash, 2010*). Furthermore, alternative patterns of

Corresponding author
Franz Goller, f.goller@utah.edu

sleep activation have been reported and led to different, debated hypotheses regarding the function of this brain activity (e.g., *Dragoi & Tonegawa, 2011*; *Dragoi & Tonegawa, 2013*; *Brawn & Margoliash, 2015*; *Eichenbaum, 2015*; *Silva, Feng & Foster, 2015*; *Grosmark & Buzsáki, 2016*).

This possibility of multiple functions of sleep activation patterns is also suggested by the occurrence of replay in well-established and stereotyped motor behaviors (*Dave & Margoliash, 2000*; *Margoliash & Schmidt, 2010*). Mechanisms for how sleep activation enhances consolidation of recently acquired memories may be different from those involved in stereotyped motor sequences, but this distinction has not been clearly made. In general, our understanding of how night-time motor replay in the brain can benefit motor performance is impeded by the fact that we do not have detailed insight into the nature of the replayed motor patterns.

One of the best postulated examples for motor replay is learned vocal behavior (*Giret, Edeline & Del Negro, 2017*). In the zebra finch, activity of neurons in sensorimotor motor (HVC) and motor cortical (the robust arcopallial nucleus, RA) areas strongly suggested that they were activated in song-like manner at night during song playback as well as spontaneously (*Dave & Margoliash, 2000*; *Hahnloser, Kozhevnikov & Fee, 2002*; *Hahnloser, Kozhevnikov & Fee, 2006*). However, constraints on recording from a large number of neurons simultaneously and on maintaining single unit responses over sufficient time spans did not permit detection of which song sequences are replayed at night. Oscine birdsong is a learned behavior, whose control involves coordination of multiple motor systems to generate a highly stereotyped acoustic sequence (*Brainard, 2008*; *Goller & Cooper, 2008*; *Suthers & Zollinger, 2008*; *Méndez et al., 2010*; *Riede & Goller, 2010*; *Beckers, 2013*). Insight into the specific motor patterns of night-time brain activation would therefore be a major advance in our understanding of how activity during sleep might benefit stereotyped motor control of complex behavior. Here we use song-like night-time activation of the muscles controlling the vocal organ of birds as a sufficiently detailed output to allow identification of which song syllable sequence is replayed in the brain during night-time.

## MATERIAL AND METHODS

During this study ten adult male zebra finches were used (age >120 days), and all procedures and experiments were performed in accordance with protocols approved by the Institutional Animal Care and Use Committee (IACUC) at the University of Utah (#16-03014). Zebra finches were housed individually in a 31.8 cm × 22.9 cm × 27.9 cm wire cage with newspaper lining. They were fed a mixture of red and white millet, canary seed, and water *ad libitum*. This diet was supplemented with peas and corn every other day. Before any surgical procedure was performed, a baseline recording of directed song was obtained using a directional microphone (Audio-Technica 835B; Audio-Technica, Machida, Tokyo, Japan) placed approximately 20 cm away from the bird. The microphone output was amplified using a Brownlee 410 amplifier (200–400×; Brownlee Precision, San Jose, CA, USA) and connected to an analog to digital converter (National Instruments, Austin, TX, USA). Recordings were obtained using Avisoft-Recorder (Avisoft Bioacoustics, Berlin, Germany) at 44.1 kHz sample rate.

Zebra finches were accustomed to the procedures three to five days before the surgery by first attaching a leash and backpack to the bird. The backpack is custom-built and consists of a Velcro tab on the back that is secured on the bird with elastic bands around the base of each wing and around the upper part of the thorax. Birds were tethered by a wire, which was fixed on the backpack and connected to a balancing lever arm positioned above the cage to allow the bird to move freely. Once birds resumed singing activity on the tether, surgical procedures were initiated. One hour prior to surgery, birds were deprived of food and water.

Surgery was performed under general anesthesia by administering Ketamine/Xylazine. During surgery, a flexible cannula was inserted into a thoracic air sac. The cannula was sutured and adhered (Vetbond; 3M Animal Care Products, St. Paul, MN, USA) to the rib cage and attached to a piezoresistive pressure transducer (Fujikura FPM-02PG; Fujikura, Tokyo, Japan) mounted on the backpack. Next, the syrinx was accessed by a skin incision in the furcula area and opening of the interclavicular air sac membrane. The tips of custom-built bipolar wire electrodes (California Fine Wire, SS 304; 25 $\mu$m) were inserted into the syringeal muscles. Bipolar electrodes were placed in two different muscles during surgery. The electrodes were adhered to the surface of the syringeal muscles with tissue adhesive (Vetbond). Electrode wires were routed subcutaneously to the backpack where the ends were attached to a connector from which larger wires made connection to signal conditioning equipment. Following surgery, the bird was placed in the cage, tethered, and allowed to recover overnight.

In each bird, we recorded from two different muscles, either on the same side or from left and right muscles (Fig. S1; Table S1). We recorded from all four intrinsic muscles, because all are activated during song. In the brown thrasher (*Toxostoma rufum*), correlations of acoustic and physiological features with EMG activity of different intrinsic muscles revealed the following picture of their respective functions: control of fundamental frequency by the ventral syringeal muscle and gating of airflow by the ventral and dorsal tracheobronchial muscles as abductor and adductor of the labia (the medial portion of the dorsal syringeal muscle was not studied). In zebra finches however, the muscles appear to be much more synergistically activated, such that for example frequency control is more complex (for review see *Goller & Riede, 2013*). We therefore recorded from all intrinsic syringeal muscles to exclude the possibility that night-time activation is specific for individual muscles or acoustic features. Because zebra finch song is also generated with asymmetrical contributions of the two independent sound generators, we recorded from muscles on both sides of the syrinx (10 recordings on each side, Table S1).

Following surgery, four channels were recorded simultaneously using Avisoft-Recorder, air sac pressure, two EMGs, and audio. EMG channels were band pass filtered (100–3,000 Hz) with a gain of 1,000×–3,000× (440; Brownlee Precision, San Jose, CA, USA). Audio recording was done with the same settings that were used to record pre-surgery song. Prior to any nighttime recording, we obtained multiple examples of complete, directed song from the male during the day by placing a female in a cage next to him. The spectrograms from these songs were compared to those recorded pre-surgery to ensure that no damage was done to the syringeal muscles during surgery.

During the night, all channels were continuously recorded at a sample rate of 8 kHz. Recordings were started and stopped manually. The photoperiod was set from 6:00 AM to 8:00 PM and was controlled by an automatic timer.

Comparison of spontaneous EMG activity to EMG activity that occurred during song was done using custom written software implementing the following procedures. The simultaneous measurement of air sac pressure and muscle activity (EMG) for singing birds was used to segment the latter into syllable-related fragments. We computed the envelopes of the EMG signals, and those smoothed time series were cut at specific time points, chosen in the following way. The minima of the pressure patterns with subatmospheric values of air sac pressure were computed, and then, for each minimum, we looked for the immediately following minimum of the simultaneously measured EMG envelope. Those times were used to cut the EMG envelope in $N$ segments, corresponding to the each syllable of the song. These EMG segments corresponding to syllables are hereafter called syllable templates.

The envelopes of EMG signals were computed by applying a Hilbert transform (impulse response filter length IRFL $= 128$), followed by a first order integration ($\tau_{\iota\nu\tau} = 0.01$), and a Savitsky Golay filter (Nleft $= 256$, Nright $= 256$). We applied this procedure to the syllable templates, as well as to the EMG data recorded during the night ($n(t)$). Then, we computed the correlation between the envelope of the night data $n(t)$ and the envelope of each of the syllable templates, shifted at delays $\tau$ between 0 and $T - T_{\text{template}}$, where $T$ is the duration of the night recording and $T_{\text{template}}$ the duration of the template analyzed. In this way, for each of the templates, we obtained a continuous time series whose largest peaks indicate a high correlation between a fragment of the envelope of the night data, and the syllable template under analysis. In order to identify the largest correlation values, we computed for each bird, all the maxima of the correlation coefficients with all the templates. Then, we fitted a Gaussian G $(\mu, \sigma)$ to the resulting histogram, and defined the threshold value $X$ such that $Z = X - \mu/\sigma = 1.64$. With this choice of threshold, every segment of the envelope of the night activity $n(t)$, whose correlation with a template is higher than $X$, was found to correspond to a segment of the time series in which activity was found (no false positives), and was similar to the template. To test the automated search algorithm, we compared EMG patterns of syllables during song to establish correlation coefficients for matching and non-matching syllables. For matching syllables the mean correlation coefficient and STD was $0.75 \pm 0.23$ ($n = 75$) and for comparisons of different syllables of $0.24 \pm 0.19$ ($n = 75$). The criterion of 0.8 for identification of EMG patterns that match specific syllables therefore constitutes a conservative approach toward identification of nocturnal EMG patterns. No false positive segments were identified with this approach, although some matching EMG segments may have been discarded with this high threshold. Visual inspection of >500 identified cases was used to further confirm the selectivity of the automated search approach and the choice of 0.8 as threshold correlation value.

To quantitatively assess night-time EMG sequences for syllable sequence, we identified the one syllable which occurred most frequently during song-like activation (SLA) for each individual. Then we quantified the highest correlation score for all other syllables around the identified syllable. To account for timing differences between song and SLA, we

searched within 640 ms near the time point of the syllable as expected from day-time song motifs. From these data we compiled histograms of correlation scores for each individual to assess syllable sequences that occur during SLA.

## RESULTS

Adult male zebra finches produce a stereotyped song sequence that is composed of a repeated series of acoustically distinct syllables, called motif (*Zann, 1996*; *Franz & Goller, 2002*; *Williams, 2008*; *Wood et al., 2013*). Each song syllable is generated by a stereotyped and characteristic respiratory pulse, which is coordinated with specific activation patterns of syringeal muscles (Figs. 1A and 1B) (*Vicario, 1991a*; *Goller & Cooper, 2004*; *Méndez et al., 2010*). After recording subsyringeal air sac pressure and electromyograms (EMG) from syringeal muscles in adult male zebra finches during song, we then monitored physiological activity during the night. Syringeal muscles showed spontaneous SLA during the night. Using a search algorithm based on template matching of the EMG patterns for song syllables, we scanned night-time files for activity. The algorithm showed sufficient distinctive power that permitted reliable identification of EMG patterns from different syllables (see 'Methods') during day-time song and night-time activity. Even with stringent criteria for the search algorithm, SLA in the syrinx was found remarkably frequently throughout the night, with a range of 110–2,370 (seven birds, over a total of 350 hrs of night-time recording) syllable-like occurrences spread throughout the night-time period (Fig. 2A). However, while SLA occurred in the syrinx, the respiratory system was not simultaneously activated in a song-like fashion and, therefore, no sound was produced. Respiratory rate during syringeal SLA ($1.59 \pm 0.46$ Hz; $n = 179$; 6 birds) was indistinguishable from normal breathing at night ($1.53 \pm 0.36$ Hz; $n = 177$; 6 birds; paired $t$-test, $p = 0.59$). Exhalation amplitude and duration were not song-like during SLA. Amplitude expressed as normalized relative voltage output during song was $8 \pm 0.23$ ($n = 83$ from 5 birds) whereas amplitude during syringeal SLA was $0.9 \pm 0.04$, $n = 100$ from 5 birds. Additionally, the duration of expiratory pressure pulses during syringeal SLA was $0.61 \pm 0.33$ s compared to $0.13 \pm 0.07$ s during song. The prolonged duration of expiration compared to awake breathing indicates that the birds were asleep (Fig. 2B). Of all the SLA events, we only found a different, more song-like respiratory pattern once in one bird, but the pressure was still not sufficiently elevated to result in phonation (Fig. 2B).

Although complete EMG patterns of the entire motif were occasionally replayed at night (Fig. 1), most of the EMG activity was not a stereotyped repetition of activity during day-time song. To quantify this, we used mean EMG patterns of individual syllables as a template to scan the night-time files for SLA occurrence and then determined how much of the full song motif was replayed. We chose data from five birds with the most distinctive EMG signals for different syllables. First, we used a subset of identified SLA occurrences to categorize 457 SLA events into one of seven categories of activity (Fig. 3). The categories describe the degree of completeness of the motif and capture deviations in timing from a typical song motif. Of these seven categories, on average over 50% of SLA occurrences fell into the single syllable or partial syllables categories. In contrast, replays of complete motifs occurred in only 7% of the total instances of SLA (Fig. 3D, Table S2).

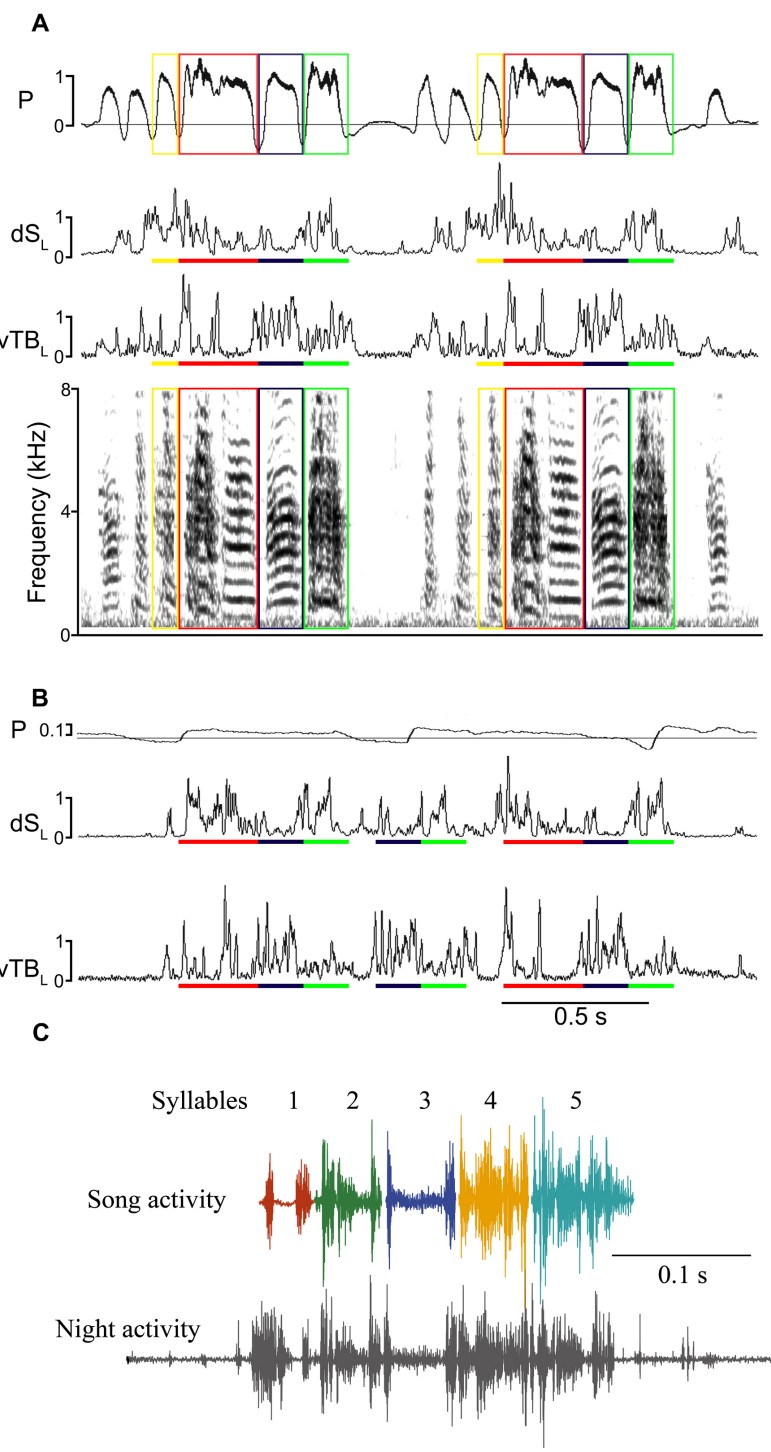

**Figure 1** **Song-like activation (SLA) of syringeal muscles occurs at night without sound generation.**
(A) Day-time song (shown spectrographically) is accompanied by a stereotyped air sac pressure pattern
(P, horizontal line illustrates ambient pressure, relative units) and EMG activity ($dS_L$ and $vTB_L$ are left
dorsal syringeal and left ventral tracheobronchial muscles). (continued on next page...)

**Figure 1 (…continued)**
Colored boxes outline individual syllables, and the corresponding colored bars indicate the EMG activity associated with that syllable. (B) An example of SLA during the night showing activation of syringeal muscles without concurrent song-like air sac pressure patterns. EMG patterns of individual syllables are identified by color bars as in (A). To illustrate the effectiveness of the algorithm for detecting similarity between song and SLA activation, correlation values for $vTB_L$ are as follows: song motif 1vs 2 (A) blue, 0.91, red, 0.85, green, 0.85. SLA vs song (A, B) blue, 0.75 and 0.71; red 0.69 and 0.64; green 0.76 and 0.79. SLA includes multiple syllable repeats of varying similarity and with missing activity for one syllable. (C) examples of syllable-specific EMG patterns of the motif of one bird (templates) with a complete motif sequence from night-time activity for comparison as identified in the automated search procedure. Complete motifs occurred infrequently during SLA.

Consistent with previous work (*Vicario, 1991a*; *Franz & Goller, 2002*), motor gestures of song production in adult male zebra finches were highly stereotyped (Figs. 4A and 4B). In contrast, night-time SLA was extremely variable when compared to stereotyped EMG activity of the song motif during day-time song (Fig. 4C). Motor replay consisted of single or multiple syllables without a discernible pattern for which particular syllable in the motif preferentially occurred during SLA. The timing between syllable EMG patterns was also more variable during night-time activity when compared to timing during song. The mean duration between syllables during night-time SLA was longer by 3 ms compared to day-time song and was generally more variable (Fig. 3E).

In addition to the variable timing and inconsistent ordering of syllables during SLA, other phenomena occurred in SLA that were never seen in stereotyped song production. EMG patterns of individual syllables showed deviations from the stereotyped pattern of day-time song. Incomplete syllables occurred, with either the first or the last half of the activity pattern missing or only present at a lower EMG amplitude. In other instances, parts of a typical EMG pattern for a syllable were omitted with the rest of the syllable pattern shifted in time, while the other muscle performed the standard song like pattern (Fig. 3C). This caused a mismatch in the SLA pattern between the two syringeal sound generators.

Second and in addition to analyzing this subset, we quantified SLA structure for the entire data set. For each bird, we chose SLA occurrences of the most frequently generated syllable and compared the EMG activity near this syllable to all other templates, allowing for different timing. If a full motif was executed, the other cross correlation coefficients should be near our cutoff threshold of 0.8. Any deviations from a full motif will yield low correlation coefficients for the other syllables. The data show clearly for all birds that the full motif was rarely produced (Fig. 4D), as correlation coefficients for other syllables rarely exceed the 0.8 cutoff criterion. Furthermore, activation patterns for different syllables occurred at very different rates, because correlation coefficients for some syllables remained much lower than those for others. Whereas the mean correlation coefficients for the target syllables ranged from 0.82 to 0.89, those for the other syllables ranged from 0.39 to 0.82. Finally, this analysis reveals differences between individuals in respect to the syllable composition of SLA sequences (Fig. 4D).

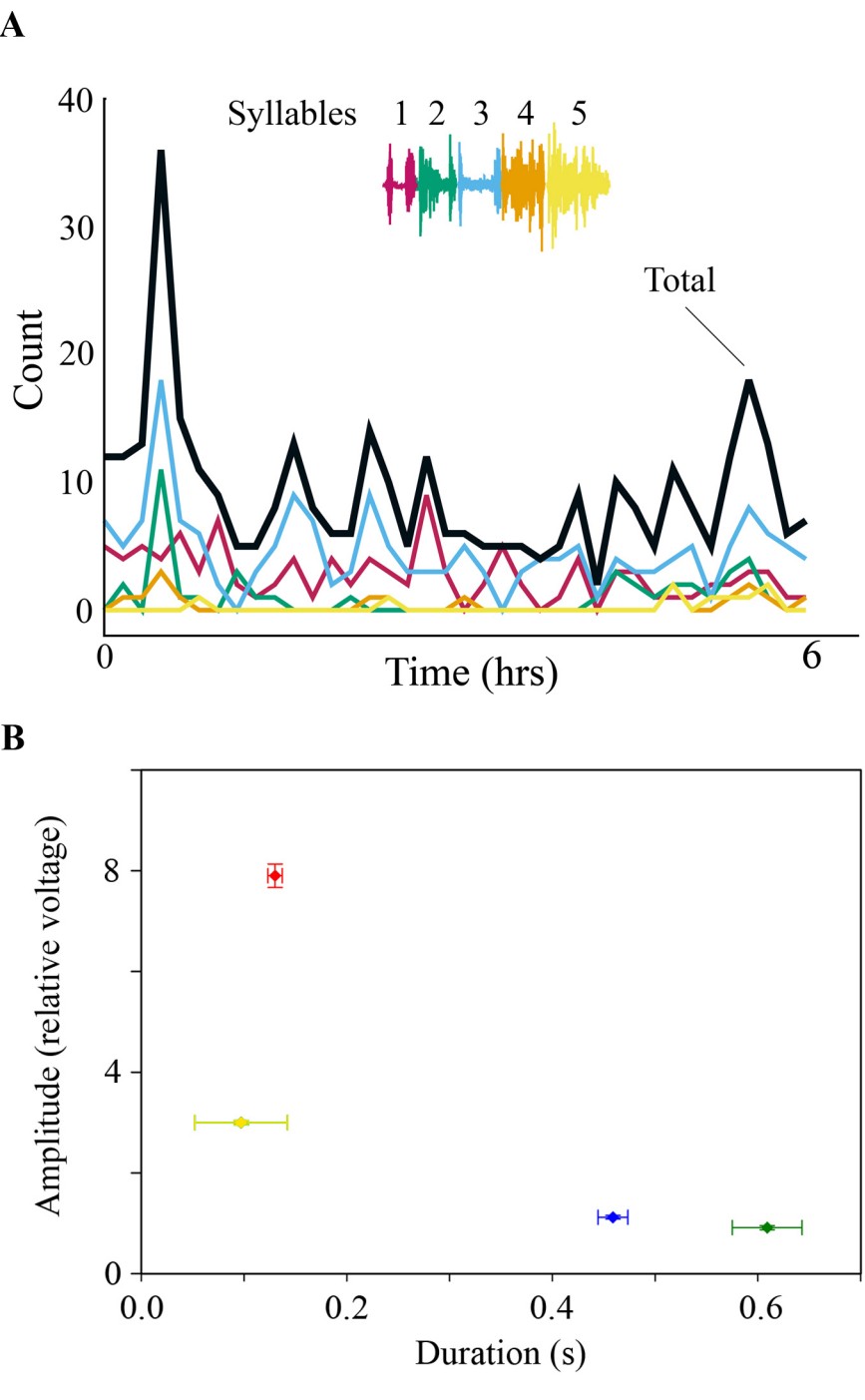

**Figure 2** (A) SLA occurred throughout the night. Specific syllables are identified by color and are re-played with varying frequency during the night. The counts are occurrences of individual syllables as determined by the algorithm detection threshold during SLA. (B) Respiration during night-time SLA (green data point) was normal quiet respiration during sleep and differed in amplitude and duration from daytime quiet respiration (blue) as well as from day-time singing (red). In only one individual night-time SLA occurred occasionally with simultaneously song-like duration and increased amplitude of expiratory pulses (yellow data point). Values are means ± 1 s.d. ($n = 250$; except for yellow data point $n = 3$).

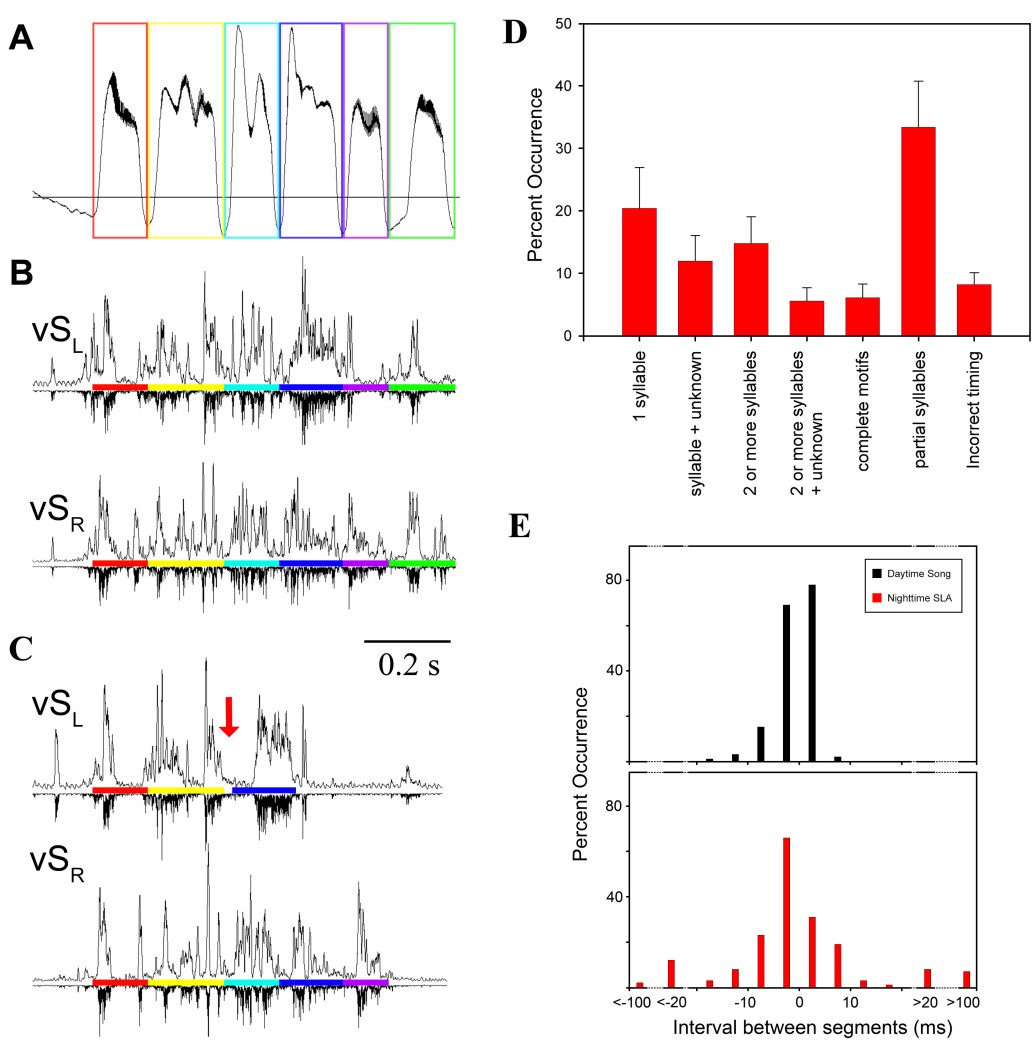

**Figure 3** **SLA showed lack of left–right coordination.** (A) Respiration during song with colored boxes outlining individual syllables. (B) EMG activity pattern in the indicated syringeal muscle during song. The upward trace shows a rectified (time constant 0.1 ms) and integrated (5 ms) EMG signal and the downward trace is a rectified raw signal for comparison. EMG patterns which correspond to a certain syllable are underlined with the respective color. (C) An example of SLA with altered syllable sequence (red arrow indicates missing syllable). (D) Occurrence of SLA patterns assigned to different categories for illustration of variability. Two or more syllables refers to incomplete motifs. Less than 7% of all occurrences were complete song motifs. (E) Inter-syllable intervals during SLA were also more variable (red) than inter-syllable intervals during song (black). To quantify inter-syllable intervals, we arbitrarily set a threshold of 200 ms on either side of a SLA pattern for it to be considered independent of another. Large negative intervals (>20 ms) in the SLA histogram indicate that the beginning of an EMG pattern for a syllable was missing with the last half shifted back. Measurements were taken as if the entire syllable pattern were present.

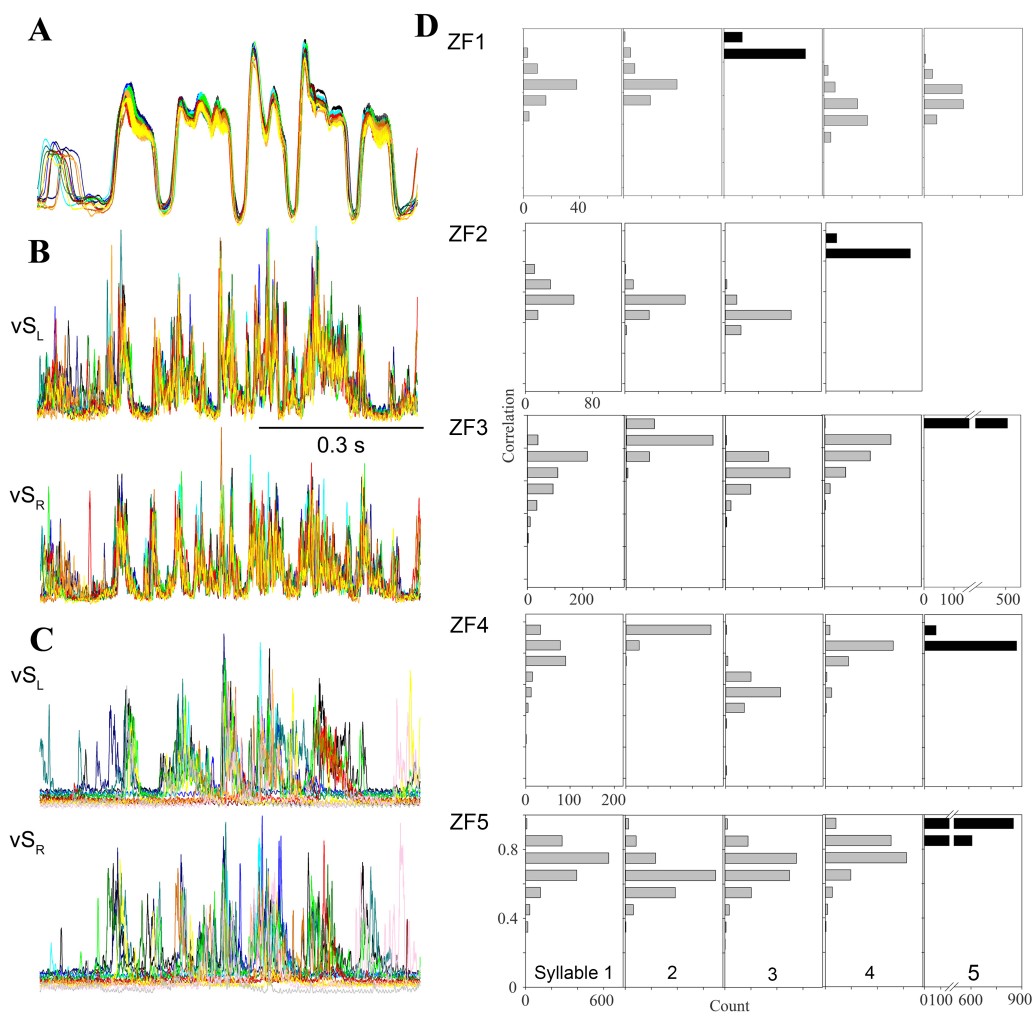

**Figure 4** **Day-time song is much more stereotyped than night-time SLA.** (A–C) Overlaid traces (different colors) of 20 song motifs illustrate the high stereotypy of song production ((A) subsyringeal air sac pressure, (B) rectified and integrated EMG of left and right ventral syringeal muscles during day-time song). (C) During SLA in contrast, EMG patterns in the same muscles are much more variable (20 different occurrences overlaid). (D) Distribution of correlation coefficients for EMG activity around the most frequently produced syllable (black bars) for each bird. The low scores for activity around the target syllable indicate that the full sequence is rarely produced and different syllable patterns are generated at different frequencies (different syllables of the 5 individuals exceeded the $r = 0.8$ threshold between 0.8 and 99.5% when the target syllable was found in SLA, $n = 10{,}104$). For example, in the first bird syllable 3 is the most frequently produced syllable, and the correlation scores for syllables 4 and 5 show that they are less frequently produced than 1 and 2.

## DISCUSSION

Here we have shown that night-time SLA does not only occur in the forebrain circuitry (HVC and RA—*Dave & Margoliash, 2000*; *Hahnloser, Kozhevnikov & Fee, 2002*; *Hahnloser, Kozhevnikov & Fee, 2006*; Area *X*—*Yanagihara & Hessler, 2012*), but that this activity also is relayed to the syringeal muscles. Whereas during song production RA initiates activation of

both respiratory pre-motor circuits and the syringeal motor neurons (nXIIts) (e.g., *Vicario, 1991b*; *Roberts et al., 2008*; *Schmidt, McLean & Goller, 2012*), during SLA only activation of the syringeal motor system occurs. The specific mechanism for disengaging respiration from the song motor sequence is not known, but inhibition at the level of the respiratory pre-motor nuclei may prevent the song-like activation of the respiratory system and, thus, sound generation.

The functional significance of activating the syringeal muscles during night-time motor replay is unclear. Because song generation involves coordinated activity of respiratory, vocal organ and upper vocal tract motor systems, the activation of only one of them during SLA is unlikely to provide useful peripheral feedback information for consolidation of the central song motor program. The activation of the whole motor circuit including the syringeal motor nucleus (nXIIts) may be necessary for providing benefits to the maintenance of the motor program. Alternatively, the activation of the syrinx may not be serving any function for song program consolidation, but, unlike respiration, may not have been selected against as it produces no externally visible movement and is likely not very energetically costly (*Oberweger & Goller, 2001*; *Franz & Goller, 2003*). Whether or not it may provide stimuli needed for the homeostatic maintenance of the superfast syringeal muscles remains unclear (*Elemans et al., 2008*; *Uchida et al., 2010*).

Nevertheless, the feed through of central activity to the syrinx provides the first detailed insight into the frequency of occurrence and specific features of motor replay of song-like activity in the brain. Such detailed information is not available for any complex behavior and, thus, enables us to characterize spontaneous motor activity for the first time. SLA occurred throughout the night period and very frequently. Replayed motor gestures for individual syllables are not consistently ordered into the correct motif sequence, are often incomplete, and even coordination of the left and right motor pathways to the two sound generators of the syrinx is not consistently present.

Zebra finch song is generated by two independently controlled sound generators, which typically contribute simultaneously to harmonic stack syllables, but high-frequency syllables are generated only on the right side, while the left side is closed to airflow (*Goller & Cooper, 2004*; *Jensen et al., 2007*). Because closure of the labia is an active process, activation of left and right muscles accompanies production of high-frequency syllables. The activation patterns of muscles therefore allow identification of all syllables, irrespective of whether the side of the syrinx contributes to sound production. The fact that some SLA in the muscles shows lack of left–right coordination points toward activation of the left and right forebrain pathways without the synchronization that is typically present during day-time song. The mechanisms for this coordination are not completely understood, but may involve cross connections in medullar, mesencephalic and diencephalic components of the song control circuit (e.g., *Schmidt, McLean & Goller, 2012*).

These findings from muscle recordings are not inconsistent with spontaneous RA activity during sleep (*Dave & Margoliash, 2000*), because the firing patterns of individual neurons in RA is such that they spike at specific syllable segments and are silent during the remaining song components. A full assessment of song-like activity during sleep is therefore not possible from these neuronal data. Whereas the spike patterns of spontaneous

activity during sleep can be matched to activity during song performance, the absence of spikes and non-matching bursts are difficult to interpret (*Dave & Margoliash, 2000*; *Shank & Margoliash, 2009*; *Rauske et al., 2010*). Syringeal activation therefore provides more information about which song elements are replayed and permits the interpretation that sleep activation is not replay of the stereotyped motor program for song. This conclusion therefore differs from the one derived from ensemble recordings in the hippocampus, where night-time activation of place cells is thought to replay previously experienced sequences fairly faithfully (e.g., *Wilson & McNaughton, 1994*; *Skaggs & McNaughton, 1996*), albeit sometimes with different tempos (e.g., *Euston, Tatsuno & McNaughton, 2007*). In contrast, in a tactile exploration task, rats showed neuronal reverberation, rather than accurate replay, in multiple forebrain areas during slow-wave sleep (e.g., *Ribeiro et al., 2004*). The motor replay patterns in adult zebra finches occur in another context (stereotyped motor program not recently learned motor behavior) and as such night-time activity constitutes a highly variable motor activation that may be similar to plasticity experienced during vocal learning (*Tchernichovski et al., 2001*; *Derégnaucourt et al., 2005*; *Shank & Margoliash, 2009*; *Ölveczky et al., 2011*).

During the sensorimotor period of song development, birds "practice" song production and generate error signals in the auditory feedback that are thought to be used for refining imitation of the acquired song model. Interestingly, SLA in HVC occurs less frequently in young birds during the sensorimotor phase than in adult birds (*Crandall et al., 2007*). In adult birds, however, song production is highly stereotyped, and variation in tempo and acoustic parameters (e.g., frequency) is very small (1–5%; e.g., *Franz & Goller, 2002*; *Cooper & Goller, 2006*; *Glaze & Troyer, 2006*; *Crandall et al., 2007*; *Williams, 2008*; *Méndez et al., 2010*; *Wood et al., 2013*). Frequent SLA and its high variability relative to the stereotyped motif of day-time song could be related to the maintenance of the motor program.

## CONCLUSIONS

Although it is currently unknown which function this high variability in SLA may serve, some possibilities are suggested here. Inspired by systems in which "preplay" has been proposed (e.g., *Dragoi & Tonegawa, 2011*; *Dragoi & Tonegawa, 2013*), motor circuits may generally explore motor space during sleep. The specific relevance of such activity for a highly stereotyped motor program, as the song production program in zebra finches, is unclear. Another possible mechanism is that variable SLA may enhance the stability of the song motor program by generating internal error signals that contribute to motor stability. An efference copy of HVC activation is thought to be involved in auditory feedback controlled song learning (e.g., *Troyer & Doupe, 2000*; *Prather et al., 2008*; *Mooney, 2009*; *Bolhuis, Okanoya & Scharff, 2010*; *Brainard & Doupe, 2013*; *Fee, 2014*). In adult birds this efference copy could predict motor output, which could internally be compared to the motor representation of stereotyped song. This internal error generation may counteract a possible decay in motor program stability in the absence of error signals (e.g., *Criscimagna-Hemminger & Shadmehr, 2008*; *Kitago et al., 2013*; *Vaswani & Shadmehr, 2013*; *Brennan & Smith, 2015*) and may facilitate the incorporation of new neurons into the song circuit

(e.g., *Alvarez-Buylla, Theelen & Nottebohm, 1988*; *Nottebohm, 2008*; *Pytte et al., 2012*) without jeopardizing the stereotypy of day-time singing. Although merely speculative at this point, this proposed model makes clear, testable predictions and generates a distinction between memory consolidation by repeating stereotyped patterns during sleep (replay) and stabilization of motor programs through internal error generation induced by variability in activation.

## ACKNOWLEDGEMENTS

We thank Natalia Verzhbitskiy for help in the laboratory.

### Funding

This research was funded by the NIH. Ezequiel Arneodo was supported by a Premio Fundación Bunge y Born postdoctoral fellowship. The funders had no role in study design, data collection and analysis, decision to publish, or preparation of the manuscript.

### Grant Disclosures

The following grant information was disclosed by the authors:
NIH.
Premio Fundación Bunge y Born postdoctoral fellowship.

### Competing Interests

The authors declare there are no competing interests.

### Author Contributions

- Brent K. Young and Franz Goller conceived and designed the experiments, performed the experiments, analyzed the data, wrote the paper, prepared figures and/or tables, reviewed drafts of the paper.
- Gabriel B. Mindlin analyzed the data, contributed reagents/materials/analysis tools, wrote the paper, prepared figures and/or tables, reviewed drafts of the paper.
- Ezequiel Arneodo analyzed the data, contributed reagents/materials/analysis tools, wrote the paper, reviewed drafts of the paper.

### Animal Ethics

The following information was supplied relating to ethical approvals (i.e., approving body and any reference numbers):
The IACUC of the University of Utah approved all procedures (# 16-03014).

### Data Availability

Goller, Franz (2017): GY63 daytime song. figshare.
https://doi.org/10.6084/m9.figshare.5406718.v1.
Goller, Franz (2017): GY63 night. figshare.
https://doi.org/10.6084/m9.figshare.5406730.v1.

## Supplemental Information

Supplemental information for this article can be found online at http://dx.doi.org/10.7717/peerj.4052#supplemental-information.

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
