# Peer review of "Adult zebra finches rehearse highly variable song patterns during sleep"

_PeerJ, doi:10.7717/peerj.4052_

## Round 0.1 · original submission · Minor Revisions

Please make sure to take into account all of the suggestions by both reviewers when you make corrections to your manuscript.

Reviewer 1 ·

Basic reporting

excellent throughout

Experimental design

query point re method is elaborated in general comments

it is an original and innovative design and research question

Validity of the findings

see below-- relates to method

Additional comments

The paper ‘Silent songs: What birds rehearse during sleep’ is both innovative and remarkable (in technique) and results and opens a window to mechanisms of control of error-coding in song learning practice in zebra finches. The paper is also beautifully written, clear in outline and outcomes and belongs to one of the best papers I have read in a while.

There are two important queries that I have for this paper-or really one major point. It concerns the syringeal muscles, or rather a missing explanation/ description in the method detailing which were used and why, ultimately affecting the results and conclusion.

1. To make the point, one can perhaps begin with a supplementary figure (Fig.S1), in principle the same as in the Larsen/Goller paper of 2002, but not as well marked in the muscle groups.
In this paper under review the caption might confuse readers.
Songbirds/oscines have at least 4 pairs of syringeal muscles—so the description saying that there are ‘at least six’ does not work well here. Why not give the exact number of pairs of syringeal muscles in the zebra finch and present this as a definite statement? In the Figure, four muscles are labelled (the largest, it says) leaving the reader wondering where the other muscles might be.
Moreover, as only three of these of these 4 identified muscles per side have been used for recording, it is not clear which ones. This is confusing for the reader. It would make it easier if a reader were not to be referred to a Table in order to figure out which of the four muscles shown in the figure were, in fact, used for recordings.

My suggestion
1) Provide and identify either all syringeal and tracheobronchial muscles of the zebra finch in this image if there are indeed more (as the caption suggests)—or the caption explains that the image showing only the largest has in fact x number more..
2) Drop the ‘at least’ in the caption and stay with the number of muscles in the finch so the reader gets to know how many muscles zebra finches actually have. Since muscle numbers vary between song birds and between oscines and sub-oscines (sub-oscines have three pairs) it is important to be told exactly how many there are in the study species.
3) It would be helpful if, in the figure itself, information was provided and identified of those muscles of which recordings were made (especially since the number in the figure is not the same as in the text)

Under methods, lines 95-97, it is stated in each bird, recordings were obtained from either 2 different muscles on the same side, or from left and right muscles (Fig. S1; Table S1). Again, this is vague- so did it vary between finches whether recordings were made using q, 2 or 3 muscles and when using both sides, were the same muscles used on either side used or different ones?

It is best to spell out these methodological details.

If I remember the important Larsen/Goller paper of 2002 correctly, the various syringeal muscles have different functions. According to an older paper by Goller and Suthers 1996, the largest syringeal muscle, m. syringealis ventralis has no gating function but in its EMG activity was positively correlated with frequency modulation and with fundamental frequency so it would make sense to me if recordings were taken from this muscle but what about the other muscles?
Can the authors please provide a rationale for the muscles chosen and what the authors hypothesised specific muscles would be able to show, (what can adductors or abductors show?).
It would also be very useful to know whether the recordings were taken equally from left and right side (laterality is a consideration). It has been shown in a number of songbirds that song is not just produced bilaterally but certain frequencies (often at the higher end) may be produced exclusively by the left side (several publications especially by Suthers, but also by Suthers,Wild and Kaplan)..

I am concerned that, if the hypothesis of why certain muscles were chosen and what they are assumed to show is not presented and discussed convincingly, readers may not believe the results. The errors may be a result of the manner in which the recordings were sampled. Equally the authors may convincingly show why any syringeal or even tracheobronchial muscle will do for recording but, either way, any such explanation and detail, in my view, would strengthen the paper considerably. I have no doubt that this is easy to fix, at least for these authors!

2. a smaller point but important
Lines 63,64—not all songbirds produce song that is highly stereotyped—but this sentence reads as if stereotyped song is universally applicable to all songbirds and, of course, that would be incorrect. The sentence simply needs and identifyer e.g., such “in zebra finches” or “in many songbirds, including zebra finches” --it’s stated clearly in lines 158-159 where it needs no addition or change.

I have no further comments to make other than to express the hope that this paper can address the method query satisfactorily. In my view, it is overall a very strong and innovative paper that should be published as soon as possible. It will be of wide interest to those working in neuroethology, psychology and in avian cognition.

Reviewer 2 ·

Basic reporting

In this paper Goller and colleagues exploit the muscle activity (EMGs) that coincides with neuronal activity of song control regions of songbirds during sleep in order to evaluate the potential meaning of this neuronal activity. The latter is thought being important for memory consolidation, and the activity of song control areas is one of the prime examples of rehearsals during sleep.
The authors computed EMGs that represent the various motor units (so called syllables) in the song of adult zebra finches. Then, they compared the envelop of the EMGs with the activities that they recorded during the night in the same muscles and animals. With this procedure they showed that most muscle activity during sleep does not correlate with the pattern found in singing birds. In particular, rarely the animals produced EMGs during the night that would represent the entire sequence of motor units (called the motif). From this, the authors conclude that it is unlikely that the birds replay the song during the sleep and conclude that the nightly neuronal activity of song control regions is not involved in song memory consolidation.

Experimental design

The methods are sound.

Validity of the findings

The methods are sound and the results are convincing. However, the discussion is somewhat disappointing and the data representation (figures and legends are somewhat careless) needs more work.

Additional comments

In general the methods are sound and the results are ok. However, the discussion is somewhat disappointing and the data representation (figures and legends are somewhat careless) needs more work. Further, the authors should replace/delete “silent songs” in/from the title. This wording is misleading since certain birds indeed produce silent songs.

Abstract:
I suggest to delete the last sentence. Further, the data are not supporting the introductory sentence of the abstract. Thus, the authors might want to adjust the abstract.

Discussion:
Lines 281-296: This entire paragraph should be re-written. The authors make first the statement that the nightly activity does not resemble the song motif EMG and as such does not help to consolidate the song motor memory. But then they state that the nightly neuronal activity might produce noise that helps to avoid the decay of the motor memory. What is the difference between consolidation and avoiding decay? Actually, this reads as if the authors have no clue about the meaning of the nightly activity. So, either they should really explain what it might mean (not just using unspecified wording) or make the statement that the function of the nightly activity is unclear. Here, the authors should critically refer to previous evidence that nightly replays are indeed replays and important for consolidation.

Figure legends:
Line 475: Include SLA.
Line 485: The authors should state that this example is a rare observation.
Line 489: explain what are the counts based on, time? Replays?
Line 503: What do you mean with timing of syllable like patterns during SLA (red)? I guess inter-syllable intervals. Please rephrase the sentence.
Lines 510-514: Numbering A-D seems wrong. Please use the same number of motifs/SLA in both cases and state that the same colors were used so that the overall color difference would be meaningful. Further, state how many replayed “syllables” were indeed significantly similar to the song syllable of these 5 animals.

Figures:
Fig. 1B: Include correlation values of the various syllables
Fig. 3C: Include vSL, vSR.
Fig. 3D: Break down “2 or more syllables” and “2 or more syllables +unknown” in order to support your point that entire motifs were rarely produced.
Fig. 3E: Occurrences: Counts, percentage? Include 80, 40 in the upper panel.
Fig.4B-C: Please adjust the number of included data sets (See above).
Fig. 4C: include vSL, VSR.
Fig. 4D: Include syllables 1-5 and animals 1-5 into the figure.

Minor points:
Line 164: Move explanation of SLA to line 148.
Lines 174/175: Include statistic for this statement.
Line 183: Move this sentence to the discussion or delete.
Line 191: What do you mean with “visually categorized 457 SLA events into one to seven categories of activity?” Aren’t you using some statistical method for this?
Line 204: Is “3 ms longer” a statistically relevant difference. If so, include STAT or delete.
Lines 252-256: This statement seems out of place at this position?
Line 258: do you mean: during singing?, please specify.
Line 266-268: Please include the evidence for night-time replay in other systems?

---

## Round 0.2 · Minor Revisions

You do not make any mention of left-right asymmetry, or absence thereof. Given the evidence of asymmetry of bird song, it would be important to mention this even if briefly.

---

## Round 0.3 · accepted · Accept

Thank you for your corrections. Your paper is very interesting and it reports on an excellent study.